# Spontaneous Postoperative Reduction of Ulnar Aneurysm by Simple Decompression of Guyon’s Canal in a Patient with Hypothenar Hammer Syndrome: A Case Report

**DOI:** 10.3390/reports7040101

**Published:** 2024-11-17

**Authors:** Ettore Gasparo, Adrian Gustar, Matteo Atzeni, Pietro Luciano Serra, Filippo Boriani

**Affiliations:** 1Plastic Surgery and Microsurgery Unit, Department of Surgical Sciences, Faculty of Medicine and Surgery, University Hospital “Duilio Casula”, University of Cagliari, 09124 Cagliari, Italy; ettoregasparo94@gmail.com (E.G.); teo.atzeni@gmail.com (M.A.); 2Plastic Surgery Unit, Department of Medical, Surgical and Experimental Sciences, Sassari University Hospital Trust, University of Sassari, 07100 Sassari, Italy; pietro_serra@hotmail.it; 3Department of Medical Sciences and Public Health, University of Cagliari, 09042 Monserrato, Italy

**Keywords:** Guyon’s canal syndrome, hypothenar hammer syndrome, ulnar artery aneurysm

## Abstract

**Background and Clinical Significance:** Guyon’s canal syndrome is a pathological condition caused by compression of the ulnar nerve at the level of the wrist. It is less frequent than other compression syndromes of the upper limb (cubital and carpal tunnel), and different causative agents, including vascular lesions, are described. Among these, aneurysm of the ulnar artery is described in the literature as an infrequent aetiology. **Case Presentation:** We report the case of a 25-year-old young man with Guyon’s canal syndrome caused by an aneurysm of the ulnar artery, who underwent surgical decompression of the Guyon’s canal without intervening on the aneurysm. The postoperative course was free of complications, and the patient reported satisfaction, with reduced symptoms. Clinical examination and ultrasound imaging showed mass reduction of the aneurysm in the postoperative period, which appears to be an evolution hitherto undocumented in the literature. **Conclusions:** Many treatments are available for Guyon’s canal syndrome. Past medical and surgical treatments, duration and severity of symptoms, causes, and pathogenesis are important for therapeutic choice. Surgical treatment based on ligament section and lysis of the Guyon’s canal downstream, without any action on the aneurysm and with ulnar artery preservation, determined a reduction in terms of volume, relief of the symptoms, and patient satisfaction. With this case we describe a surgical therapeutic option for the treatment of Guyon’s canal syndrome caused by an aneurysm of the ulnar artery, in which surgery is limited to canal decompression and consequential aneurysm mass reduction with concomitant relief of symptoms.

## 1. Introduction and Clinical Significance

Originating at levels C8-T1, the ulnar nerve is a branch of the brachial plexus and travels through Guyon’s canal to supply motor and sensory innervation to the hand and fingers. Compression injury of the ulnar nerve occurs less often in Guyon’s canal than at the medial epicondyle of the humerus, known as the cubital tunnel.

The boundaries of Guyon’s canal include its anatomic “roof”, the palmar or volar carpal ligament, and its “floor”, the flexor retinaculum or transverse carpal ligament. The transverse carpal ligament runs deep to Guyon’s canal to the ulnar side (medially) at the level of the wrist while its radial (lateral) portion runs superficial to the median nerve, forming the “roof” of the carpal tunnel.

The pisiform bone, piso-hamate ligament, and abductor of the small digit represent the ulnar confine, while the hook of the hamate (hamulus) represents the radial confine.

There are many potential deleterious stimuli to the distal ulnar nerve, including compression, flogosis, injury, or vascular insufficiency. Aetiologies include ganglion cyst; the hook of hamate fracture/displacement (fracture of the hook of hamate); tumours (lipoma); repetitive injury; aberrant muscle syndrome (e.g., abductor digiti minimi) or overabundance of fat tissue in the canal; or thrombus on the ulnar artery or aneurysm (e.g., hypothenar hammer syndrome, a rare occupational or recreational malady resulting from repetitive microtrauma to the ulnar artery at the level of Guyon’s canal) [1,2]. Guyon’s canal syndrome is a less common peripheral ulnar neuropathy caused by damage to the distal part of the ulnar nerve as it passes through a narrow anatomical tunnel at the wrist and is usually treated through canal decompression and possible therapy of the associated lesion. With this case report, we describe a case of aneurysm-associated ulnar nerve compression at the Guyon’s canal successfully treated with simple decompression through canal decompression.

## 2. Case Presentation

A patient, 25 years old and right hand dominant, came to the plastic surgery department showing symptoms that suggested Guyon’s canal syndrome.

The patient was a manual worker who was suffering from an aneurysm in the volar region of the left hand of ulnar artery of approximately 19 mm at maximum diameter, between the deep subcutaneous tissue and the muscle of the hypothenar eminence, with the presence of a 4 mm thrombus on the posterior wall, confirmed both by ultrasound and MRI. No features of ischemia, necrosis, or distal embolism were detected. He had no reported history of disease, allergies, or G6PDH deficiency; he was on drug treatment with low molecular weight heparin, and he was a smoker.

In his work history, he reported being a gutter worker.

The patient reported chronic pain in the left hand with paraesthesia localized predominantly to the volar region of the left hand and the third, fourth, and fifth fingers.

On clinical examination, the patient had positive Tinel’s sign and Froment’s sign.

The electromyography performed showed slowed conduction velocity along the ulnar nerve.

Pain, discomfort, and paraesthesia were present for over 1.5 years. Ultrasound imaging allowed for the diagnosis of an ulnar artery aneurysm of approximately 19 mm (Figure 1).

The treatment consisted of a volar incision to the palm under regional anaesthesia, sedation, and arm tourniquet, followed by section of the volar carpal ligament and dissection of the ulnar fascicle to the depth and distal part of the Guyon’s canal, with separation of the aneurysmatic artery from the nerve (Figure 2).

The patient was discharged after one day of hospitalization with the recommendation to carry out ultrasound scans in the following months and with post op heparin therapy. The postoperative course was characterized by immediate improvement of neurological symptoms (reduction of pain and improvement of sensitivity).

In particular, two-month postoperative ultrasonography proved the reduction of the aneurysm from 19 mm to 16.3 mm (Figure 3), and three-month postoperative ultrasonography indicated a further reduction to 16 mm (Figure 4). Six months after the procedure, the patient was completely pain free and with full sensory recovery in the ulnar distribution of the hand.

It is likely that, based on the previous disease course, a missed decompression or delayed treatment of the cause of Guyon’s canal syndrome would undoubtedly have led to worsening symptoms, paraesthesia, decreased grip strength, progressive atrophy of the intrinsic muscles of the hand with loss of the adduction–abduction ability of the fingers (especially the little finger), difficulty in holding objects, as well as a probable increase in the volume of the aneurysm itself and thus a possible increase in the volume of the thrombus within it.

## 3. Discussion

Our proposed hypothesis is that manual labour, with chronic work of repetitive percussion on the hook of the hamate of the left hand, gradually caused inflammation that determined chronic oedema, with stagnation of fluids that led to fibrosis with narrowing of the Guyon’s canal. This had repercussions on the ulnar artery that narrowed downstream, causing an aneurysmal dilation upstream, thus leading to the formation of this aneurysm over time.

In the literature, there are many medical and surgical therapies proposed for the treatment of hypothenar hammer syndrome (due to an aneurysm of ulnar artery), although there are no definitive guidelines for the treatment of this condition.

The literature describes conservative treatments that can be used for patients who do not have a risk of necrosis or ischemia. These consist of adjusting activities; quitting smoking; managing pain; using calcium channel blockers, ⍺-blockers, β-blockers, heparin, and steroids; and intravenous and oral vasodilators, such as prostaglandins and prostacyclin, to reduce sympathetic tone and vasospasm [3]. More invasive treatments include stellate ganglion block or intra-arterial administration of reserpine [4]. Non-surgical therapy utilizes a percutaneous method that guides intra-arterial thrombolytics to the location of the ulnar artery thrombosis. This treatment option involves administering a recombinant tissue plasminogen activator or urokinase through a catheter for thrombolysis. These intra-arterial treatments are less invasive, but they often do not resolve the neurological symptoms underlying Guyon’s canal syndrome. Furthermore, they are not without complications; in particular, potential issues from using thrombolytic therapy include hematoma and bleeding at the site of access [3].

Regarding surgical treatment, some techniques that can be used are intraarterial thrombectomy or aneurysm resection with or without artery reconstruction, with vein or arterial graft, or the simple ligation of ulnar artery. Zied et al. [5] reported a case of a young man diagnosed with Guyon’s canal syndrome caused by an ulnar artery aneurysm, who underwent surgical decompression. The surgical treatment consisted of opening and releasing the roof of the Guyon’s canal with removal of the aneurysm and good post-operative results. However, this approach resulted in the sacrifice of the ulnar artery, which was ligated, with a consequent reduction in vascular flow to the hand.

Another treatment proposed for hypothenar hammer syndrome due to an aneurysm of ulnar artery is the simple ligation of the ulnar artery [6].

Regarding the reconstruction options, a different surgical approach was discussed by Hui-chou HG et al. [7] through resection of the abnormal arterial segment with ligation of vessel or reconstruction based on venous [8] or arterial grafts [9]. However, this kind of surgery requires an expert microsurgery team (which is not always available at a plastic surgery department) and/or surgical robot [10] and may be subject to complications, like numbness in the donor site, neuroma formation, graft failure, thrombosis [11], and hypertrophic scar formation.

Thus, why not perform a repair of the aneurysm and/or thrombectomy or bypass using vein or artery grafts? In the present case, the cause of the aneurysm was an ab extrinsic and not ab intrinsic cause. In particular, there was no real weakening of the wall as in classic primary aneurysms, but rather it was an aneurysmal dilatation secondary to the fact that there was compression upstream. We considered it excessive to perform such techniques and instead placed confidence in a restitutio ad integrum of the vessel wall once the artery was decompressed. Our experience is encouraging given the significant reduction in size after only 3 months and improvement in symptoms after 6 months. Ulnar artery sacrifice by ligation was never considered by our team.

In fact, our goal was to save the artery by performing a simple decompression at the level of the roof of Guyon’s canal through the section of the piso-hamate ligament and volar carpal ligament in such a way as to cause a diminished downstream compression with reduction of pressure and therefore shrinking of the aneurysm itself, thus saving the ulnar artery (in order to reduce postoperative complications and especially preserve hand and finger function) (Figure 2).

The small 4 mm thrombus within the aneurysm was treated conservatively through the prescription of cycles of heparin.

## 4. Conclusions

In conclusion, the authors propose a conservative surgery for the treatment of Guyon’s canal syndrome caused by an ulnar artery aneurysm through a simple decompression of the roof of the canal downstream with a reduction of the upstream pressure, thereby obtaining a significant shrinkage of the aneurysm, as proven by ultrasonography.

In fact, with this technique, there was a progressive and significant improvement in the clinical symptoms without sacrificing the ulnar artery, which is an important artery for the vascularization of the hand, the sacrifice of which can lead not only to a deficit in the perfusion of the hand and fingers (acutely) but also to alterations in sensitivity and problems of thermoregulation, mild muscle atrophy, and ischemic pain (chronically).

We can hypothesize that as the size of the aneurysm is reduced by the decompression carried out, the venturi effect, which is responsible for the change in blood flow velocity and thus the formation of the thrombus within the aneurysm, may be reduced. The progressive reduction in the volume of the aneurysm thus leads us to assume that the size of the thrombus will not increase. In fact, the patient we treated had no sign or symptom attributable to distal ischemia and embolism at 3 months post-surgery.

## Figures and Tables

**Figure 1 reports-07-00101-f001:**
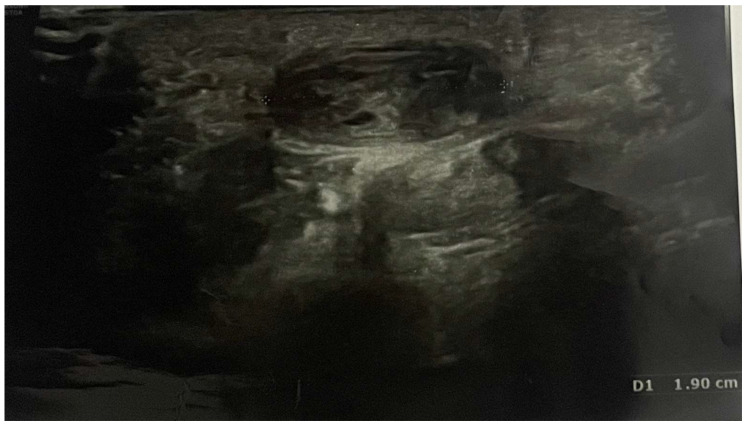
Preoperative ultrasonography; D1 shows the length of ulnar artery aneurysm (19 mm).

**Figure 2 reports-07-00101-f002:**
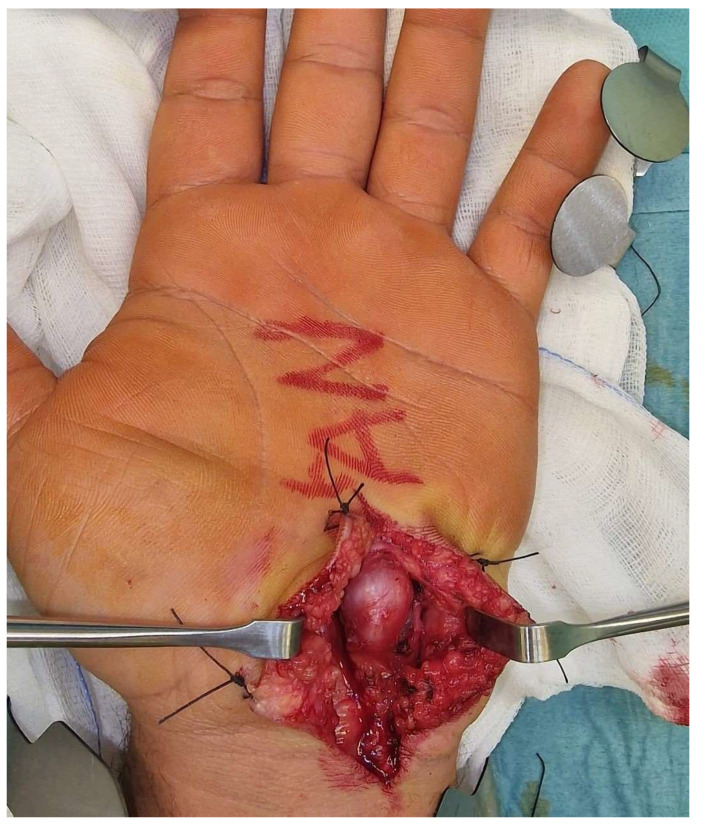
Intraoperatory view; the aneurysm of the ulnar artery was identified. Dissection of the subcutaneous planes to expose the ulnar nerve while preserving the ulnar nerve integrity. Complete section of the volar carpal ligament and the most distal segment of the piso-hamate ligament.

**Figure 3 reports-07-00101-f003:**
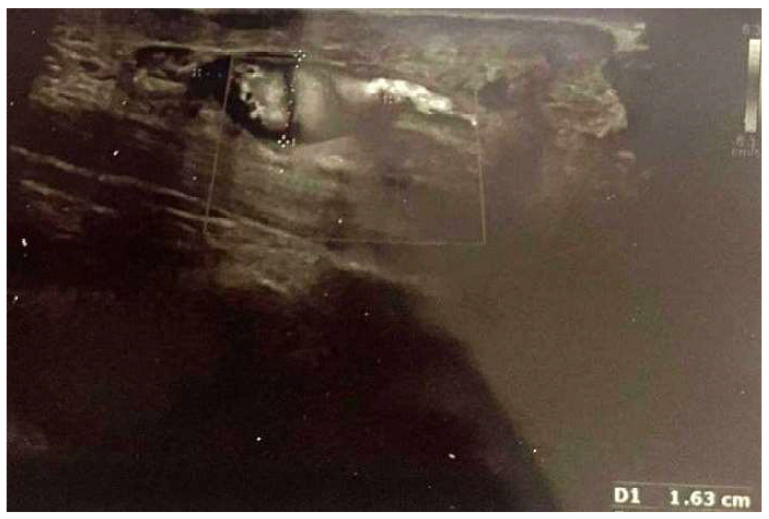
Two-month postoperative ultrasonography shows a reduction of ulnar artery aneurysm from 19 mm to 16.3 mm.

**Figure 4 reports-07-00101-f004:**
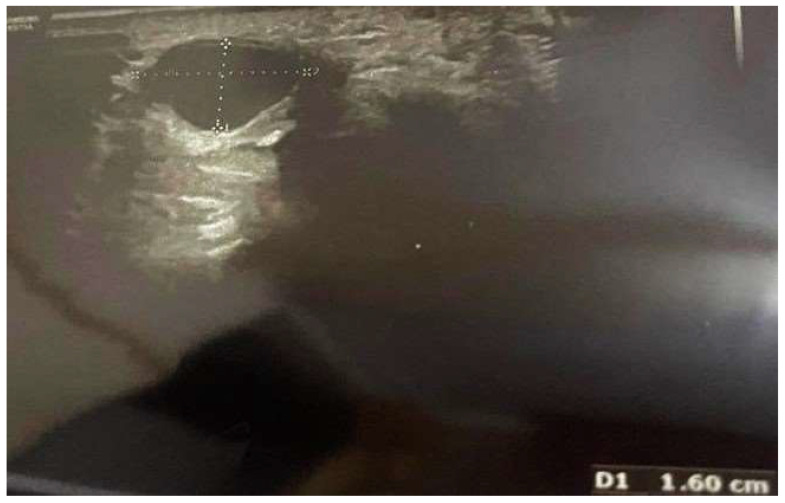
Three-month postoperative ultrasonography shows a reduction of ulnar artery aneurysm from 16.3 mm to 16.0 mm.

## Data Availability

The study includes the original data. Interested parties can contact the corresponding author for more information. The data are not publicly available due to privacy concerns.

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
