# Peer review of "Spontaneous Postoperative Reduction of Ulnar Aneurysm by Simple Decompression of Guyon’s Canal in a Patient with Hypothenar Hammer Syndrome: A Case Report"

_reports, 2024, doi:10.3390/reports7040101_

Round 1

Reviewer 1 Report

Comments and Suggestions for Authors

Well written manuscript of hypothenar hammer syndrome with conservative treatment of an ulnar artery aneurysm.

Comments:
1) what were the presenting symptoms for this patient?

2) preop imaging revealed intra-arterial thrombus in a 19 mm aneurysm, were there any features of distal embolism?

3) after decompression, what was the reason to NOT perform a repair of the aneurysm and thrombectomy?

4) in your case, did you consider anticoagulation post op to avoid distal embolization?
5) discussion and conclusion lack clinical evidence to support your proposal, please consider discussing the standard of care for ulnar artery aneurysms with thrombus. Your approach is very atypical and places the surgery team at medical legal risk is the patient embolizes after the decompression without repairing the aneurysm

6) nice images were included

Author Response

Response to reviewer 1

Well written manuscript of hypothenar hammer syndrome with conservative treatment of an ulnar artery aneurysm.

Dear Reviewer 1, many thanks for your compliments.

Comments:
1) what were the presenting symptoms for this patient?

As is written in the manuscript: “The patient reported chronic pain in the left hand with paresthesia localized predominantly to the volar region of the left hand, and the 3rd, 4th, and 5th fingers. Pain, discomfort and paresthesia had been present for over 1.5 years.” More details were added in the text.

2) preop imaging revealed intra-arterial thrombus in a 19 mm aneurysm, were there any features of distal embolism?

Indeed, the pre-operative ultrasound showed the presence of a 4 mm thrombus within the aneurysm. The patient had been suffering from this pathology for over a year and a half, but had never had any ischemic disturbances distally. This was extensively assessed through the pre-operative clinical examination and through instrumental investigations (which also assessed the vascularization of the fingers), and no signs of embolism distal to the aneurysm were detected. This aspect was added to the manuscript text, case presentation section.

3) after decompression, what was the reason to NOT perform a repair of the aneurysm and thrombectomy?

About the treatment of a thrombus at the level of an ulnar artery aneurysm, the literature is very controversial in this regard. In particular, the cause of the aneurysm on this patient was an ab extrinseco and not ab intrinseco cause, in fact there was no real weakening of the vascular wall as in classic primary aneurysms, but it was rather an aneurysmal dilatation secondary to the fact that there was compression upstream.

Based on these considerations we hypothesized that by reducing the compression upstream, the aneurysm would reduce in size causing an improvement in symptoms. We did not perform an aneurysm repair and thrombectomy because we did not want to subject the patient to a high surgical stress and above all because the thrombus was very small, and the patient was undergoing heparin therapy. These concepts were added in the discussion section.

4) in your case, did you consider anticoagulation post op to avoid distal embolization?

Yes, the patient underwent pre- and post-operative heparin therapy

5) discussion and conclusion lack clinical evidence to support your proposal, please consider discussing the standard of care for ulnar artery aneurysms with thrombus. Your approach is very atypical and places the surgery team at medical legal risk is the patient embolizes after the decompression without repairing the aneurysm

Thank you, the literature is unclear regarding the treatment of HHS in the presence of thrombotic occlusion. Treatment options range from conservative approaches through the use of vasodilators, prostaglandin steroids, and heparin, to more invasive treatments such as stellate ganglion blockade or intra-arterial administration of reserpine, to surgical procedures such as intrarterial thrombectomy or aneurysm resection with or without artery reconstruction.

However, in case of asymptomatic thrombotic partial occlusion (from a vascular point of view) with therefore no signs of distal ischemia, no treatment is necessary. If, on the other hand, signs of occlusion begin to appear, conservative or minimally invasive therapy can be initiated up to surgical procedures if the clinical picture worsens.

The patient we treated, as mentioned above, had no type of signs or symptoms attributable to distal ischemia, and he was already undergoing heparin therapy, so we afforded to treat the cause of the aneurysm (ab extrinseco compression) to improve the neurological symptomatology, which instead was the most important pathological condition for the patient to treat at that time.

Obviously, if the patient should start to manifest any kind of symptomatology referable to a thrombotic occlusion, we will consider undergoing surgery to treat the thrombus.

6) nice images were included

Thanks for that, we really appreciate

Reviewer 2 Report

Comments and Suggestions for Authors

This is a well written manuscript presenting an intersting case report.

Introduction

It is useful to emphasize what makes this case particularly rare or noteworthy mentioning the potential consequences of delayed intervention, linking back to why this case report is clinically valuable.

There are minor gramatical and stylistic correction to be suggested:

Line 64: the sentence "He did not report any pathology, no allergies and no G6PDH deficiency" could be streamlined to "He had no reported history of disease, allergies, or G6PDH deficiency."

Line 77: "Caracterized" should be corrected to "characterized."

Line 82: The sentence “full sensation recovery to the ulnar area of his hand” is better as “full sensory recovery in the ulnar distribution of the hand.”

Discussion

Some phrases are repeated in the discussion and conclusion. For example, the impact of preserving the ulnar artery on vascularization is stated multiple times. Please condense these statements.

"over the time" is better as "over time".

Please structure clearly the presentation of treatments: first, discuss conservative options, followed by nonoperative approaches, and conclude with surgical treatments.

It is useful ncluding a brief mention of other surgical alternatives (e.g., bypass grafts or reconstructive options) with an explanation as to why these were not chosen in your case and instead of merely listing treatments, explicitly highlight how the proposed treatment aligns or contrasts with prior cases and techniques. For example, after mentioning Zied et al.'s approach, comment on how preserving the ulnar artery may reduce post-operative complications and preserve hand function.  Including a brief mention of other surgical alternatives (e.g., bypass grafts or reconstructive options) with an explanation as to why these were not chosen in your case could enhance the rigor of the discussion.

References appropriate

Figures good quality.

  •  
  •  

Author Response

Response to reviewer 2

This is a well written manuscript presenting an intersting case report.

Introduction

It is useful to emphasize what makes this case particularly rare or noteworthy mentioning the potential consequences of delayed intervention, linking back to why this case report is clinically valuable.

Thanks for the insights, we have now added this important concept in the introduction section.

There are minor gramatical and stylistic correction to be suggested:

Line 64: the sentence "He did not report any pathology, no allergies and no G6PDH deficiency" could be streamlined to "He had no reported history of disease, allergies, or G6PDH deficiency."

Line 77: "Caracterized" should be corrected to "characterized."

Line 82: The sentence “full sensation recovery to the ulnar area of his hand” is better as “full sensory recovery in the ulnar distribution of the hand.”

Thanks for spotting this. Corrections have been made.

Discussion

Some phrases are repeated in the discussion and conclusion. For example, the impact of preserving the ulnar artery on vascularization is stated multiple times. Please condense these statements.

"over the time" is better as "over time".

Please structure clearly the presentation of treatments: first, discuss conservative options, followed by nonoperative approaches, and conclude with surgical treatments.

Thank you for pointing this out, sentences have now been reordered accordingly.

It is useful ncluding a brief mention of other surgical alternatives (e.g., bypass grafts or reconstructive options) with an explanation as to why these were not chosen in your case and instead of merely listing treatments, explicitly highlight how the proposed treatment aligns or contrasts with prior cases and techniques.

Thank you, these techniques have been now mentioned more in details. They had been poorly described in this case report because the literature is unclear regarding the treatment of an ulnar artery aneurysm with a small thrombus within it, which gives no clinical symptoms of distal ischemia.

Since the patient was on heparin therapy, we preferred to treat the painful and paresthetic neurological symptomatology at the level of the fingers caused by Guyon's canal syndrome.

the technique we performed, however, does not contrast possible vascular surgical therapy in the future, if the thrombus shows signs of distal ischemia. It is currently totally asymptomatic.

For example, after mentioning Zied et al.'s approach, comment on how preserving the ulnar artery may reduce post-operative complications and preserve hand function.  Including a brief mention of other surgical alternatives (e.g., bypass grafts or reconstructive options) with an explanation as to why these were not chosen in your case could enhance the rigor of the discussion.

Thank you, these inputs will definitely be incorporated

References appropriate

Thanks

Figures good quality.

Thank you

Round 2

Reviewer 2 Report

Comments and Suggestions for Authors

The manuscript was improved like as suggested and now is ready to be published